



# Measurement Report: MAX-DOAS measurements characterise Central London ozone pollution episodes during 2022 heatwaves

Robert G. Ryan[1*], Eloise A. Marais[1], Eleanor Gershenson-Smith[1], Robbie Ramsay[2], Jan-Peter Muller[3], Jan-Lukas Tirpitz[4], Udo Friess[5]

[1]Department of Geography, University College London, London, UK
[2]Natural Environment Research Council Field Spectroscopy Facility, Edinburgh, UK
[3]Mullard Space Science Laboratory, Department of Space and Climate Physics, University College London, Holmbury St Mary, UK
[4]Airyx GmbH, Eppelheim, Germany
[5]Institute of Environmental Physics, Heidelberg, Germany
* Now at: School of Earth Sciences, University of Melbourne, Australia

*Correspondence to*: Eloise A. Marais (e.marais@ucl.ac.uk)

**Abstract.** Heatwaves are a substantial health threat in the UK, exacerbated by co-occurrence of ozone pollution episodes. Here we report on first use of retrieved vertical profiles of nitrogen dioxide ($NO_2$) and formaldehyde (HCHO) over Central London from a newly installed Multi-Axis Differential Optical Absorption Spectroscopy (MAX-DOAS) instrument coincident with two of three heatwaves for the hottest summer on record. We evaluate space-based sensor observations routinely used to quantify temporal changes in air pollution and precursor emissions over London. Collocated daily mean tropospheric column densities from the high spatial resolution space-based TROPOspheric Monitoring Instrument (TROPOMI) and MAX-DOAS, after accounting for differences in vertical sensitivities, are temporally consistent for $NO_2$ and HCHO (both R = 0.71). TROPOMI $NO_2$ is 27-31% less than MAX-DOAS $NO_2$, as expected from horizontal dilution of $NO_2$ by TROPOMI pixels in polluted cities. TROPOMI HCHO is 20% more than MAX-DOAS HCHO; greater than differences in past validation studies, but within the range of systematic errors in the MAX-DOAS retrieval. The MAX-DOAS lowest layer (~55 m altitude) retrievals have similar day-to-day and hourly variability to the surface sites for comparison of $NO_2$ (R ≥ 0.7) and for MAX-DOAS HCHO versus surface site isoprene (R > 0.6) that oxidizes to HCHO in prompt and high yields. Daytime ozone production, diagnosed with MAX-DOAS HCHO-to-$NO_2$ tropospheric vertical column ratios, is mostly limited by availability of volatile organic compounds (VOCs), except on heatwave days. Temperature dependent biogenic VOC emissions of isoprene increase exponentially, resulting in ozone concentrations that exceed the regulatory standard for ozone and cause non-compliance at urban background sites in Central London. Locations in Central London heavily influenced by traffic remain in compliance, but this is likely to change with stricter controls on vehicle emissions of $NO_x$ and higher likelihood of heatwave frequency, severity and persistence due to anthropogenic climate change.



## 1 Introduction

Heatwaves in the UK cause ozone pollution episodes that worsen heat-related premature mortality (Doherty et al., 2009; Johnson et al., 2005; Pattenden et al., 2010; Rooney et al., 1998; Stedman, 2004). In summer 2022, London experienced three heatwaves, declared when surface air temperatures in Greater London exceed 28°C for at least three consecutive days (McCarthy et al., 2019). The first heatwave in June (15th-17th) was unusually early (McCabe, 2022); in the July heatwave (16th-19th) London temperatures exceeded 40°C for the first time in century-long temperature measurement records (Kendon, 2022); and more non-COVID related excess deaths were registered during the August heatwave (11th-15th) than the more intense heatwave in July (ONS and UKHSA, 2022). The extreme heatwave in July was due to an exceptionally high pressure system and clear conditions causing a so-called "heat dome" over the UK (Kendon, 2022). There was also a surge in residential fires during this heatwave due to ideal ignition conditions following a sustained and intense drought in southeast England (London Fire Brigade, 2022).

Ozone is a secondary pollutant formed from photochemical reaction of nitrogen oxides ($NO_x \equiv NO + NO_2$) and volatile organic compounds (VOCs). Concentrations of ozone are typically low in London, as cold and cloudy conditions dominate, and $NO_x$ emitted by vehicles titrate ozone via its reaction with nitric oxide (NO) (AQEG, 2009). Due to large traffic sources of $NO_x$, ozone production in London is for most of the year limited by availability of VOCs (Jin et al., 2017). This is despite decline in $NO_x$ emissions in Central London of ~3.3% $a^{-1}$, according to the London Atmospheric Emission Inventory (Mayor of London, 2021). During heatwaves, increases in surface ozone result from a combination of downwelling of ozone-rich air from the free troposphere, advection of polluted air from continental Europe, faster kinetics from warm temperatures and sunshine, and large enhancement in emissions of the reactive biogenic VOC isoprene due to exponential dependence of emissions on temperature (Lee et al., 2006; Sillman and Samson, 1995).

In mid-June 2022, a Multi-Axis Differential Optical Absorption Spectroscopy (MAX-DOAS) instrument was installed on the rooftop of an 11-story building at the University College London (UCL) Bloomsbury campus, providing measurements during the July and August heatwaves. These are passive UV/visible instruments that measure direct and scattered sunlight by conducting discrete vertical and horizontal scans of the atmosphere (Hönninger et al., 2004). The instrument was deployed to provide long-term tropospheric observations of the vertical distribution and column integrated concentrations of UV/visible active chemicals over Central London. This includes nitrogen dioxide ($NO_2$), a regulated air pollutant and constraint on precursor emissions of $NO_x$ (Martin et al., 2003), and formaldehyde (HCHO), a prompt and high-yield oxidation product of isoprene and ubiquitous oxidation product of almost all other VOCs (Millet et al., 2006).

The MAX-DOAS instrument at UCL also adds a UK site to the existing extensive global network of MAX-DOAS instruments used to evaluate space-based single daily overpass UV/visible instruments (De Smedt et al., 2021; Pinardi et al., 2020; van



Geffen et al., 2022) and the anticipated Sentinel-4 geostationary instrument that will provide hourly daytime observations over
Europe and North Africa (Timmermans et al., 2019). Satellite observations have been vital for understanding air quality over
Greater London, in particular trends quantified from the 15+ year record of observations from the Ozone Monitoring
Instrument (OMI) used to assess the impact of emission control measures on air quality and to evaluate the accuracy of bottom-
up emissions inventories (Pope et al., 2018; 2022; Vohra et al., 2021). The recently launched TROPOspheric Monitoring
Instrument (TROPOMI) achieves higher spatial resolution than OMI and its other predecessors resolving column
concentrations over Central London. Still, TROPOMI is limited to a single piece of vertical information in the troposphere
and to a midday snapshot of the atmosphere.

Here we exploit coincidence of the MAX-DOAS instrument with the July and August 2022 heatwaves to evaluate TROPOMI
observations of HCHO and $NO_2$ over Central London, characterize ozone pollution episodes during these heatwaves, and
assess the efficacy of snapshot measurements from single overpass space-based instruments on diagnosing ozone production
over Central London.

## 2 Methods

### 2.1 Instrument Location and Viewing Geometry

A SkySpec MAX-DOAS instrument (model SkySpec-2D-210, Airyx GmbH, Germany) has been installed at the 60 m altitude
rooftop laboratory on the Bloomsbury campus of UCL at 51.52°N and 0.13°E (Figure 1) since mid-June 2022. The instrument
has UV (300-410 nm) and visible (410-556 nm) spectrometers each with 0.6 nm spectral resolution. It is fitted with a
temperature sensor that measures ambient temperature every minute and two webcams that image the sky overhead every 8
minutes. During the measurement period, the UCL MAX-DOAS was configured to measure spectra over a set of elevation
angles of 1°, 2°, 3°, 5°, 10°, 20°, 40° and 90° every 8 minutes to obtain vertically resolved information from the boundary
layer to the free troposphere.

The instrument also samples horizontally by scanning discrete viewing azimuth angles. The horizon to the north and west is
obstructed by rooftop infrastructure. To the south and east, tall buildings along the London skyline may obscure scans at low
elevation angles, necessitating that we identify azimuth angles with an unobstructed view of the horizon. On a cloud-free day
(8 July), we scanned the horizon from 90° to 181° at 14:15-15:00 UTC using the horizon scan setting of the instrument. This
measures spectra in 1° azimuth increments faster and at fewer elevations angles (1°, 2°, 4°, 90°) than the standard measurement
setting. Figure 2 shows colour indices or spectral intensity ratios at 330 nm to 404 nm ($I_{330}/I_{404}$) for 1° and 90° elevation angles.
We use this ratio, as the light intensity in the visible is sensitive to changes in sky colour resulting from interception by
buildings and the ratio of the two normalizes for atmospheric variability. Such an approach has been used to infer the presence
of clouds along the instrument viewing path (Gielen et al., 2014; Ryan et al., 2018; Wagner et al., 2014; Wagner et al., 2016).



The variability in $I_{330}/I_{404}$ at 90° elevation angle, due only to changes in light intensity over the time and horizon sampled, is < 0.03. Given this, we identify azimuth angle windows at 1° elevation angle with $I_{330}/I_{404}$ variability < 0.03 and select the centres of these (112°, 132°, and 175°; Figure 1). We use these from 9 July onward and the azimuth angles selected prior to optimisation (135°, 180°) for 1-8 July.

**2.2 Vertical Profile Retrieval**

MAX-DOAS retrievals follow two major steps to estimate vertical concentrations of trace gases. Column concentrations along the viewing path at each elevation angle (differential slant column densities or dSCDs) are first obtained using the DOAS Intelligent System (DOASIS) proprietary software (Kraus, 2006). DOASIS is founded on a long heritage of MAX-DOAS retrieval algorithm development first described by Platt and Stutz (2008). The software corrects the raw spectra for dark current, electronic offset and stray light and convolves spectral cross sections of the analysed trace gases with the slit function of the instrument. Optimized wavelength ranges are 338-370 nm for $NO_2$ and the $O_2$-$O_2$ dimer ($O_4$), as recommended following a recent MAX-DOAS intercomparison campaign (Kreher et al., 2020), and 324.5-359 nm for HCHO, as this yields lower relative dSCD fit errors than the other commonly used fit range of 336-359 nm (Ryan et al., 2020b). $O_4$ is used to constrain aerosol impacts on the atmospheric light path. Ozone also absorbs in the UV, but MAX-DOAS observations of ozone include large interference from the stratosphere that is challenging to remove (Wang et al., 2018). The DOASIS dSCD retrieval uses a $3^{rd}$ order polynomial with a $1^{st}$ order offset to correct for broadband extinction by Rayleigh scattering and instrumental features such as spectrometer stray light. Also included in the dSCD fit are terms to account for the Ring effect (Grainger and Ring, 1962) and absorption by other trace gases in the $NO_2$ and HCHO fitting windows. Further details of the DOAS fit parameters used by DOASIS are in Kreher et al. (2020).

We determine the detection limits (*DL*s) of individual dSCDs for each trace gas as follows:

$$DL = 2 \times RMS/\sigma_{max}, \tag{1}$$

where $\sigma_{max}$ is the maximum value of the absorption cross section of each trace gas and *RMS* is the root mean square of the fit residuals (Peters et al., 2012). Values of $\sigma_{max}$ are $1.0 \times 10^{-42}$ cm$^5$ molecule$^{-2}$ for $O_4$ (Finkenzeller and Volkamer, 2022), $8.4 \times 10^{-19}$ cm$^2$ molecule$^{-1}$ for $NO_2$ (Vandaele et al., 1998), and $1.3 \times 10^{-19}$ cm$^2$ molecule$^{-1}$ for HCHO (Chance and Orphal, 2011).

The second retrieval step estimates vertical profiles of aerosol extinction, $NO_2$ and HCHO using the recently developed Retrieval of Atmospheric Parameters from Spectroscopic Observations using DOAS Instruments (RAPSODI) algorithm (Tirpitz, 2021; Tirpitz et al., 2022). Compared to predecessor algorithms, RAPSODI retrieves multiple species at different wavelengths simultaneously in a shared model atmosphere. In so doing, it accounts for cross-correlations and synergistic information that improves inversion accuracy. RAPSODI uses optimal estimation with the Vector Linearized Discrete Ordinate Radiative Transfer (VLIDORT) model (Spurr, 2008; Spurr, 2006) as the forward model. Vertical profiles are retrieved on a



grid that includes 25 layers extending to 8 km with a vertical resolution that decreases with altitude from 50 m in the lowest layer to 1 km in the top layer.

Optimal estimation is ill-constrained and so requires an initial guess (a priori) of the vertical distribution of aerosols, $NO_2$ and HCHO to determine the maximum likelihood of the atmospheric state by minimizing a cost function, given the observed dSCDs at each elevation angle (Rodgers, 2000). Instead of using fixed a priori profiles of $NO_2$ and HCHO, the default option in RAPSODI, we simulate hourly mean profiles with the GEOS-Chem chemical transport model version 13.0.0

(https://doi.org/10.5281/zenodo.4618180; accessed 12 December 2021) nested over Greater London (49.25°N–59.50°N, 9.375°W–3.75°E) at 0.25° × 0.3125° horizontal resolution. Three-hourly boundary conditions are from a global simulation at 4° × 5° resolution. The model is driven with NASA GEOS-FP assimilated meteorology and uses anthropogenic emissions from the UK National Atmospheric Emission Inventory as detailed in Marais et al. (2021a) and Kelly et al. (2023). Natural emissions of biogenic VOCs, precursors of HCHO, are from the Model of Emissions of Gases and Aerosols from Nature

(MEGAN) version 2.1 (Guenther et al., 2012). Model vertical profiles are sampled from the gridbox coincident with the MAX-DOAS location (gridbox centre of 51.5°N, 0.0°E) and interpolated onto the MAX-DOAS vertical retrieval grid. To capture diurnal variability in $NO_2$ and HCHO and dampen influence of day-to-day variability in the model on the retrieval, a priori profiles are averaged into hourly monthly means for July, August and September. We test the effect of using dynamic a priori profiles from GEOS-Chem on the final retrieved values by comparison to vertical profiles obtained with fixed a priori profiles

provided with RAPSODI. The latter are exponential profiles with surface concentrations of 3.9 ppbv for both $NO_2$ and HCHO that decay with altitude with a scale height of 1 km. Surface daytime values from GEOS-Chem on the RAPSODI grid range from 1.6-10.4 ppbv for $NO_2$ and 0.6-1.2 ppbv for HCHO.

The influence of aerosols on light is determined by RAPSODI using the Henyey-Greenstein approximation for Mie scattering,

as this offers a computationally efficient alternative to explicitly invoking Mie theory (Tirpitz et al., 2022). Due to availability of measurements of aerosol optical depth (AOD) at a long-term NASA AERONET site at Bayfordbury Observatory, we use a climatology of AOD to derive a priori aerosol extinction vertical profiles rather than using GEOS-Chem. The Bayfordbury Observatory is 35 km north of UCL. There is also an instrument measuring AOD as part of the AERONET network on the UCL rooftop, but the record is too short-term and intermittent (2009, 2010, 2021, 2022) to be representative of a climatological

mean. We find that at Bayfordbury daily mean AODs at 340 nm are consistent with AOD at UCL for coincident observations from July 2021 to July 2022 (Pearson's correlation coefficient, R, of 0.94 and mean difference <10%), supporting its use. We derive aerosol extinction profiles assuming a typical exponential decline in aerosol extinction with altitude determined using a scale height of 1 km (Tirpitz et al., 2022) and multiyear (2011-2022) mean AOD of 0.23. We assume a priori uncertainties of 75% for $NO_2$ and HCHO and 50% for aerosols. The a priori correlation coefficient between each layer decreases

exponentially with vertical separation, assuming a scale length of 1 km.





The retrieval also requires knowledge of surface albedo at the site. For this, we use monthly mean surface albedo climatology from the MODerate resolution Imaging Spectroradiometer (MODIS) gridded level 3 500 m resolution UV/visible band (300-500 nm) product. We sample data for July-September over London using the NASA EarthData subsetting tool (ORNL DAAC, 165    2022). Values are 0.060±0.007 for July-August and 0.061±0.005 for September.

The optimal estimation solution includes an averaging kernel matrix that provides a measure of the sensitivity of the retrieved profile to the true atmospheric state in each layer. The trace of the averaging kernel matrix is the degrees of freedom for signal (DOFs) or the number of independent pieces of information in the retrieved profile. We identify MAX-DOAS vertical column 170    retrievals with limited information from the observations (dSCDs) as those with DOFs <1 and discard these. Very few data points (2%) are removed with this filter.

**2.3 Qualitative Cloud Detection**

Clouds may induce errors in the retrieval, as these alter the atmospheric light path over the frequently cloudy London sky. We use the $I_{330}/I_{404}$ colour index to qualitatively identify cloudy scenes, as has been done before (Gielen et al., 2014; Ryan et al., 175    2018; Wagner et al., 2014; Wagner et al., 2016). We first determine the relationship between $I_{330}/I_{404}$ and solar zenith angle (SZA) on cloud-free days in each month by visual inspection of the camera images. These include 16 and 18 July and 7 and 10-13 August. There was no completely cloud-free day in September, so we were limited to using measurements on the 17[th] when there were scattered clouds. Figure 3(a) shows the relationship between colour indices and SZA for individual observations on the selected days at 112° azimuth angle and 20° elevation angle. We use 20° as it samples more representative 180    whole sky conditions than lower elevation angles. We calculate separate 3[rd] order polynomial fits for the morning and afternoon in each month, as the azimuth angle is not due south, so the relationship between $I_{330}/I_{404}$ and SZA is asymmetric across the day. Colour indices in the morning of 17 September deviate from the polynomial fit, due to the presence of scattered clouds.

In Figure 3(b) we show sensitivity of $I_{330}/I_{404}$ to clouds over Central London by comparing a cloud-free and a cloudy day in 185    August to the Figure 3(a) fit for August. Values on the cloud-free day (11[th]) are well within 10% of the fit, as expected, as this is one of five days in August used to derive the fit. Cloudy skies on the 15[th], confirmed with the camera images, deviate by more than 10% from the fit. We identify clouds as colour indices that differ from the cloud-free fit by at least 10% and assess the influence on HCHO and NO₂ profile retrievals. This approach will be least effective at detecting clouds in the early morning and late afternoon when the intensity at 404 nm is weak, as is apparent in Figure 3(b).

**2.4 Collocated Satellite and Surface Air Quality Observations**

Global observations of tropospheric columns of NO₂ and total columns of HCHO are available from the space-based nadir-viewing UV/visible TROPOMI instrument on the Sentinel-5P satellite. Sentinel-5P was launched into low-Earth orbit on 13 October 2017 and passes overhead each day at about 13:30 local solar time (LST). TROPOMI has a ground pixel size of 5.5



× 3.5 km at nadir (Verhoelst et al., 2021) and a swath width of 2600 km resulting in daily global coverage. We use the offline
(OFFL) data products for $NO_2$ (v2.03.01) and HCHO (v2.04.01). MAX-DOAS trace gas retrievals are sampled within 1.5 h
of the satellite overpass time.

TROPOMI pixel centres are commonly sampled 0.2° around the geographic coordinates of MAX-DOAS instruments for
intercomparison of the two (Marais et al., 2021b; Pinardi et al., 2020; Ryan et al., 2020b). Instead, we use a location roughly
halfway between the MAX-DOAS instrument and the visible horizon to account for its southeast viewing direction (Figure 1).
The visible horizon at wavelengths relevant to HCHO and $NO_2$ varies between 10 and 15 km (Ortega et al., 2015), so we select
a location 6 km from the MAX-DOAS site along the central 132° azimuth at 51.49°N and 0.07°E and sample TROPOMI pixel
centres 0.2° (~20 km) around this point. This location falls within the GEOS-Chem grid sampled for a prioris in the MAX-
DOAS retrievals (Section 2.2). We average MAX-DOAS vertical profiles at all viewing azimuth angles, account for
differences in vertical sensitivity between the two instruments by smoothing MAX-DOAS vertical profiles with the TROPOMI
averaging kernels (Dimitropoulou et al., 2020; Rodgers and Connor, 2003), and integrate the smoothed profiles to calculate
MAX-DOAS vertical columns. MAX-DOAS profiles inherit TROPOMI values from above the MAX-DOAS retrieval ceiling
(8 km) due to this smoothing. We rely on data quality flags in the TROPOMI data products to filter for cloudy scenes. Data
with quality assurance flag < 0.75 are removed for $NO_2$ and with quality assurance flag < 0.5 for HCHO. This removes scenes
with cloud radiance fraction ≥ 0.5 and poor quality retrievals (De Smedt et al., 2021; Verhoelst et al., 2021).

There are no permanent surface air quality sites within 10 km of the MAX-DOAS instrument along its line of sight, but there
are four sites within 6 km of the instrument location. These are Westminster, Bloomsbury, North Kensington and Marylebone
Road (Figure 1). All are part of the national Automatic Urban and Rural Network (AURN) and all except Marylebone Road
are classified as urban background sites. The Westminster site is in a mixed commercial and residential district 17 m from the
nearest road, Bloomsbury is surrounded by a congested 2-lane road, and North Kensington is in the grounds of a school 5 m
from a quiet residential road. Marylebone Road, an urban traffic site 1 m from a frequently congested 6-lane road, also
measures the HCHO precursor VOC isoprene as part of the UK Automatic Hydrocarbon network. Measurements are hourly
means of $NO_2$ at all urban background sites, ozone at Bloomsbury and North Kensington, and isoprene at Marylebone Road.
All data were downloaded from the UK Air Information Resource website (https://uk-air.defra.gov.uk/data/, last accessed
21/11/2022). $NO_2$ is also available at Marylebone Road, but mean $NO_2$ in July-September 2022 at this site is 2.5 times more
than the mean of the other sites due to large local traffic $NO_x$ emissions. We compare the surface air quality measurements to
MAX-DOAS HCHO and $NO_2$ mixing ratios in the lowest retrieval layer (0-110 m) averaged over all three azimuth angles
(Figure 1).



## 3 Results and Discussion

### 3.1 MAX-DOAS Differential Slant Column Density Retrievals

Figure 4 shows a sample of DOASIS retrieved dSCDs of $O_4$, $NO_2$ and HCHO on 18 July 2022 at elevation angles below 90°. The most recent 90° (zenith) spectrum serves as a reference (Leser et al., 2003). Hourly variations in dSCDs are a function of atmospheric light path length and trace gas concentration. Longer light paths at lower elevation angles and in the morning and evening cause larger dSCDs in all three trace gases. Greater abundance of these trace gases in the boundary layer contribute to decline in dSCDs with elevation angle. $NO_2$ peaks in the morning and late afternoon and is at a minimum at midday at all elevation angles below 40°, whereas HCHO is relatively constant at all elevation angles. We interpret diurnal variability in MAX-DOAS vertical column retrievals of $NO_2$ and HCHO in Section 3.3. Uncertainties in the dSCDs calculated by DOASIS are relatively small (<5%) for all three trace gases. This is as expected for urban MAX-DOAS $NO_2$ and $O_4$ (Dimitropoulou et al., 2020; Ortega et al., 2015), but lower than is typical for urban MAX-DOAS HCHO (Benavent et al., 2019; Heckel et al., 2005; Ryan et al., 2020b). This may be because HCHO dSCDs in Central London are three times larger than those over Melbourne, Australia in Ryan et al. (2020b) and due to differences in HCHO wavelength ranges used by Heckel et al. (2005) and Benavent et al. (2019).

Also in Figure 4 are *DL*s of each trace gas at 1° elevation (Equation (1)). *RMS* values are typically on the order $4 \times 10^{-5}$. The *DL*s in Figure 4 represent maximum values, as there is lower signal (less UV light) at low elevations leading to relatively large residual *RMS* values. The larger relative uncertainties in HCHO dSCDs lead to relatively large *DL*s for HCHO, evident in Figure 4(c) in the early morning and late afternoon when relatively weak light intensity causes larger *RMS* values. The mean *DL*s are $9.3 \times 10^{38}$ molecules$^2$ cm$^{-5}$ for $O_4$, $1.1 \times 10^{15}$ molecules cm$^{-2}$ for $NO_2$, and $6.5 \times 10^{15}$ molecules cm$^{-2}$ for HCHO. All dSCDs on 18 July exceed the *DL*s and this is typical of the other days in the measurement period.

### 3.2 MAX-DOAS Comparison to TROPOMI

Figure 5 compares coincident MAX-DOAS and TROPOMI $NO_2$ and HCHO tropospheric vertical column densities. TROPOMI sensitivity peaks in the upper troposphere, whereas MAX-DOAS sensitivity typically peaks at or near the surface (De Smedt et al., 2021; Dimitropoulou et al., 2020). As a result of these differences in sensitivity and because $NO_2$ and HCHO concentrations peak in the boundary layer over polluted cities, smoothing MAX-DOAS with the TROPOMI averaging kernels decreases the MAX-DOAS columns by ~26% for $NO_2$ and ~48% for HCHO. The greater decline in MAX-DOAS HCHO than $NO_2$ suggests weaker sensitivity of TROPOMI to boundary layer HCHO than $NO_2$ over Central London. The information content of TROPOMI is also limited to one piece of vertical information. DOFS for TROPOMI are typically ~1 for $NO_2$ and HCHO compared to DOFS of ~3 for MAX-DOAS $NO_2$ and ~2 for MAX-DOAS HCHO for coincident observations in Figure 5. The TROPOMI cloud detection and retrieval quality flag filtering (Section 2.4) removes 43 of the 92 days in the comparison period for $NO_2$ and 31 for HCHO. Larger errors in TROPOMI HCHO than TROPOMI $NO_2$ daily means, obtained by adding



reported retrieval uncertainties in quadrature, are due to relatively large uncertainty in individual columns (on average ~6 × $10^{15}$ molecules cm$^{-2}$) and fewer coincident pixels each day for HCHO (typically 18) than NO$_2$ (typically 48).

The two instruments are temporally consistent for NO$_2$ and HCHO (R = 0.71 for both). Both also capture the steep increase in HCHO in the August heatwave of 43% from 6 to 11 August for MAX-DOAS and 65% for TROPOMI. The increase is likely due to an increase in biogenic emissions with warming that we investigate further with the surface air quality network measurements in the next section. During the comparison period, TROPOMI is on median 31% less than MAX-DOAS for NO$_2$ and 20% more than MAX-DOAS for HCHO. The regression statistics indicate that the difference is because TROPOMI

exhibits less variance in NO$_2$ than MAX-DOAS (slope = 0.73 ± 0.12) and more variance in HCHO than MAX-DOAS (slope = 1.30 ± 0.15). The statistics are relatively unaffected by increasing the 3-h sampling time window to 5 h. A robust assessment of the effect of narrowing the sampling extent of TROPOMI to 0.1°, as in Pinardi et al. (2020), is not possible, as our measurement period is brief and so will be affected by a decline in the number of coincident days from 49 to 38 for NO$_2$ and 61 to 45 for HCHO.


TROPOMI low bias in NO$_2$ is consistent with other global and regional intercomparison studies over cities (Chan et al., 2020; Dimitropoulou et al., 2020; Verhoelst et al., 2021; Wang et al., 2020). These studies used earlier versions of the TROPOMI NO$_2$ data product that underestimates cloud top pressure of low-altitude clouds, contributing to a low bias in NO$_2$ over polluted regions. This is addressed in the version we use and, as shown by van Geffen et al. (2022), reduces the negative bias in

comparison to the global network of MAX-DOAS instruments over urban areas from 32% to 23% (Lambert et al., 2021; van Geffen et al., 2022). The difference over Central London is more consistent with the earlier product versions, although this is sensitive to the two MAX-DOAS columns on 3 August and 27 September that are 2-times more than TROPOMI. The difference after excluding these points improves to -27%. In Central London, mean NO$_2$ at road traffic sites are almost 3-times more than mean NO$_2$ at urban background sites (Harrison et al., 2021), so horizontal dilution of NO$_2$ by TROPOMI pixels

likely exacerbates the discrepancies between TROPOMI and MAX-DOAS (Pinardi et al., 2020).

The difference in HCHO over Central London is opposite in sign and larger in magnitude than the median bias from global comparison studies of -10% (Chan et al., 2020; De Smedt et al., 2021). Only the site at Madrid exhibited a positive bias of 10% (De Smedt et al., 2021); less than the 20 % we obtain over Central London. The cause for a larger difference in TROPOMI

and MAX-DOAS HCHO over Central London is challenging to isolate, as the TROPOMI data version we use has updated radiances used in the slant column retrieval and an updated background correction for addressing systematic offsets (De Smedt et al., 2022). Also, the MAX-DOAS HCHO data at the sites in the De Smedt et al. (2021) global comparison employ different retrieval algorithms. According to a MAX-DOAS intercomparison study, HCHO retrieval differences can account for systematic errors in dSCDs of up to 20% (Pinardi et al., 2013).




**3.3 MAX-DOAS Comparison to Surface Air Quality Monitors**

Figure 6 shows daytime daily mean observations at the MAX-DOAS and surface monitoring sites for July-September 2022.
MAX-DOAS values are the lowest retrieved layer, so represent average mixing ratios across 0-110 m altitude centred at 55 m.
Clouds are a large source of error in retrieval of trace gas column densities from space-based instruments (Millet et al., 2006).
Previous tests have identified that clouds affect MAX-DOAS retrieval of AOD (Gielen et al., 2014; Wagner et al., 2014;
Wagner et al., 2016), but there is no equivalent assessment for trace gases. We evaluate the effect on MAX-DOAS retrievals
by averaging cloud-free observations identified with colour indices ($I_{330}/I_{404}$) at 20° elevation angles that deviate by at least
10% from the cloud-free fits in Figure 3(a) (Section 2.3). This removes almost 60% of retrieved $NO_2$ and HCHO and the
difference in July-September mean all-sky and cloudy lowest-layer mixing ratios and vertical column densities of both trace
gases is at most 3%. The limited influence of clouds is likely because of the strong sensitivity of the retrieval to the
observations.

Informed by the weak sensitivity of trace gas retrievals to clouds, we use all-sky MAX-DOAS trace gas concentrations in the
lowest layer with profile DOFs > 1 (Section 2.2). We also find that retrieved trace gas profiles are relatively insensitive to the
choice of a priori. Retrievals of vertical column densities and lowest-layer mixing ratios of both $NO_2$ and HCHO at 175°
azimuth angle in July, for example, differ by < 10% using the default static RAPSODI a priori profile versus July average
hourly mean a priori profiles from GEOS-Chem (Section 2.2), as most retrieval information is from the observations.

In Figure 6(a), daytime MAX-DOAS and surface site $NO_2$ exhibit consistent day-to-day (R = 0.85) and hourly (R = 0.69; not
shown) variability. Daytime mean $NO_2$ is 7.4 ± 3.0 ppbv at the surface and 5.4 ± 2.3 ppbv at ~55 m, the centre altitude of the
MAX-DOAS lowest layer. Based on this, the vertical gradient in $NO_2$ is -36 pptv m$^{-1}$. An autumn multiyear (October 2006
and October-November 2007) campaign in London calculated 24-h mean $NO_2$ of 22 ± 12 ppbv at the Northern Kensington
site (Figure 1) and measured 24-h mean $NO_2$ of 17 ± 9 ppbv with an in situ instrument installed at 160 m on the British
Telecommunications (BT) Tower located ~500 m west of the MAX-DOAS site (Harrison et al., 2012). This yields a similar
vertical gradient of -31 pptv m$^{-1}$, despite differences in the magnitude of surface $NO_2$ resulting from differing averaging periods
(24-h vs daytime, autumn vs summer, 2006/2007 vs 2022) that influence $NO_2$ lifetime and precursor emissions. Decline in
surface concentrations of $NO_2$ in Greater London from 2006/2007 to 2022 is ~43%, based on the 2.5% a$^{-1}$ decline in
tropospheric column densities of $NO_2$ from OMI, a reasonable proxy for trends in surface $NO_2$ (Vohra et al., 2021). The vertical
gradient of 24-h mean GEOS-Chem $NO_2$ in July-September 2022 from the lowest layer centred at 58 m to the layer above
centred at 123 m is weaker than both at -17 pptv m$^{-1}$.


MAX-DOAS HCHO is compared to surface measurements of isoprene concentrations, as HCHO is a prompt, high-yield
oxidation product of isoprene in locations with elevated $NO_x$ such as Central London (Marais et al., 2012) and in situ



observations of HCHO in Central London are limited to short-term intermittent field campaigns. Daytime isoprene averages 0.17 ± 0.13 ppbv in July-September (Figure 6(b)) and includes emissions from trees and anthropogenic sources (mostly vehicles) in London (Valach et al., 2015). Biogenic emissions are enhanced in summer due to warm temperatures, sunlight, and large leaves (Guenther et al., 1995; 2006). The contribution of biogenic sources to isoprene in summer in London ranges from 50% at the Marylebone Road traffic site in Central London to 90% at the Eltham suburban background site 16 km southeast of the MAX-DOAS site (Khan et al., 2018; von Schneidemesser et al., 2011). Abundant tree species in London that are high isoprene emitters include oak and sycamore (Greater London Authority Environment Team, 2021). July-September surface isoprene and MAX-DOAS HCHO have similar day-to-day (R = 0.78) and hourly (R = 0.62; not shown) variability. The correlation in daytime daily means is weaker in September (R = 0.59) than the other months (R = 0.76), due to decline in isoprene emissions resulting from cooler temperatures and shorter days. This may have been exacerbated in 2022 by early senescence of trees across southeast England due to a sustained drought (Rosane, 2022).

Figure 7 shows July-September average hourly mean MAX-DOAS and surface site $NO_2$ and MAX-DOAS HCHO and surface site isoprene. The lowest-layer diurnal variability is similar to the integrated column (0-8 km) for HCHO and $NO_2$. The lowest layer is on average 13% of the integrated column for $NO_2$ and 6% for HCHO. Surface site $NO_2$ peaks at night when there is no photolytic loss of $NO_2$ and during morning and afternoon rush hours. The magnitude of the morning peak is similar (9 ppbv at 7 am) for MAX-DOAS and the surface sites, whereas MAX-DOAS is less than the surface sites during the midday minimum (by 54 %) and the afternoon traffic peak (by 40 %). This suggests that the vertical gradient between the MAX-DOAS lowest retrieval layer and the surface sites, averaging -36 pptv m$^{-1}$ throughout the day, evolves from negligible in the early morning to about -50 pptv m$^{-1}$ by late afternoon due to an increase in the efficiency of photolytic loss of $NO_2$.

Diurnal variability of isoprene and HCHO differ, despite consistent day-to-day variability (Figure 6(b)). Isoprene concentrations are at a minimum (< 0.1 ppbv) at night and maximum (0.25 ppbv) at 2pm due to exponential dependence of isoprene emissions on temperature (Guenther et al., 1995; 2006; Valach et al., 2015). HCHO diurnal variability is relatively flat, increasing by 16% from 5 am to 7 pm. The relatively constant daytime HCHO over London is due to balance of sources and sinks; the latter dominated by photolysis (Marais et al., 2012). In the morning and late afternoon HCHO is mostly from oxidation of anthropogenic VOCs that have accumulated overnight or that are emitted by vehicles during rush-hour traffic as unburned hydrocarbons (Valach et al., 2015). At midday, isoprene emissions and photolysis rates peak (Valach et al., 2015). The same flat HCHO diurnal shape has been reported for summer ground-based column measurements in Beijing (De Smedt et al., 2015; Stavrakou et al., 2015). HCHO over other cities (Uccle in Belgium, Melbourne in Australia, Seoul in South Korea) has a distinct early afternoon peak during summer (De Smedt et al., 2015; Ryan et al., 2020a; Spinei et al., 2018; Stavrakou et al., 2015), reflecting larger contribution of biogenic sources (Leuchner et al., 2016; Vigouroux et al., 2018; Xiaoyan et al., 2010). The MAX-DOAS site in Melbourne, for example, is in a suburban area with lower traffic density than Central London and near high isoprene emitting Eucalyptus forests (Ryan et al., 2020a).



## 3.4 Heatwaves and Ozone Pollution in Summer 2022

Figure 6 also includes day-to-day variability in the ratio of MAX-DOAS vertical column densities of HCHO and $NO_2$ ($HCHO:NO_2$). These range from ~0.2 to ~2.8. The range of this ratio for the lowest MAX-DOAS retrieval layer is narrower

(~0.1 to ~1.0), due to greater free tropospheric contribution to the total tropospheric column for HCHO than for $NO_2$ (Section 3.2). $HCHO:NO_2$ decrease over the period examined, as cooler temperatures and shorter days lead to decline in isoprene emissions and increase in $NO_2$ lifetime and abundance. Values of $HCHO:NO_2$ are often used to diagnose whether ozone production depends on VOCs or $NO_x$ for informing policy measures to address ozone pollution (Jin et al., 2017; Ryan et al., 2020a; Vohra et al., 2021; Xue et al., 2022). Typically, ozone production regimes are diagnosed as $NO_x$-saturated or limited

by the availability of VOCs at $HCHO:NO_2 < 1$ and $NO_x$-sensitive at $HCHO:NO_2 > 2$ (Duncan et al., 2010). Ozone production in warm months in London is gradually transitioning to the $NO_x$-sensitive regime, based on trends inferred from OMI for 2005 to 2015 and attributed to $NO_x$ emission controls (Jin et al., 2017). The exact values that define ozone production regimes depends on the oxidative state of the atmosphere, and so should ideally be calibrated to local conditions (Souri et al., 2020). We use the threshold values from Duncan et al. (2010) as an approximate interpretation of ozone production. According to the

daytime daily means in Figure 7(c), ozone production in July-September 2022 is mostly $NO_x$-saturated, despite continued decline in $NO_x$ emissions in London since 2015 (Mayor of London, 2021); the last year of the Jin et al. (2017) trend analysis. During the July heatwave, daytime daily means occupy the upper end of the transition from $NO_x$-saturated to $NO_x$-sensitive.

Figure 8 shows hourly variability in the observations during the July heatwave. The maximum air temperature at 60 m of 38°C

is on 18 July and is ~2°C cooler than the maximum surface air temperature recorded in London during the heatwave. Due to very stable high-pressure conditions (Kendon, 2022), nocturnal accumulation of surface $NO_2$ leads to morning concentrations on 18-19 July that are up to 30 ppbv more than the July-September mean morning peak (Figure 7(a)). Surface $NO_2$ on the morning of the 19[th] is just below the World Health Organization (WHO) 2021 guideline for short-term (24-h mean) exposure to $NO_2$ of 25 μg m[-3] (equivalent to ~48 ppbv).


Isoprene measurements are missing from 5 pm 18 July to 2 pm 19 July. The remaining observations in Figure 8 suggest substantial increase in isoprene emissions during the two hottest days in July. Isoprene emissions have a well-known exponential dependence on temperature that is parameterized in models like the widely used MEGAN with current and recent past air temperature. We find with the temperature parameterization from Guenther et al. (2006) that the 10°C increase in air

temperature from the start of the heatwave on 17 July to the hottest hour on 19 July may have caused a 3-fold increase in isoprene emissions. As a result of the large increase in isoprene emissions, diurnal variability of HCHO on 18 July peaks at midday, consistent with the largest contribution from biogenic sources. HCHO is enhanced throughout 19 July, likely due to overnight accumulation of HCHO from the previous day and from local fires. Large fires also occurred in France and Spain due to severe and sustained hot and dry conditions (Copernicus Atmosphere Monitoring Service, 2022; Henley and Jones,



2022; Imbach et al., 2022). The Copernicus Atmosphere Monitoring Service forecast of AOD (Benedetti et al., 2009; Morcrette et al., 2009) places smoke plumes from fires in Spain and France over London from the morning of 17 July until the afternoon of 19 July. This likely led to the 0.1-0.2 km$^{-1}$ enhancements in aerosol extinction above background levels retrieved by the MAX-DOAS at 1-2 km altitude from 11am on 18 July. There was no corresponding enhancement in $NO_2$ and HCHO in this altitude range.


The large enhancements in morning $NO_2$ and associated ozone depletion during the July heatwave in Figure 8 mask severe ozone pollution and large hourly variability in the ozone production HCHO:$NO_2$ diagnostic in the daytime means in Figure 6. On 17-19 July anticorrelation of peak $NO_2$ in the morning and peak HCHO at midday leads to hourly HCHO:$NO_2$ in Figure 8(c) that ranges from 1 (ozone production transition regime) to 4.4 (strongly $NO_x$ sensitive). Lowest layer HCHO:$NO_2$ ranges

from 0.02 to 2.4 over the same period. On these days, the large increase in isoprene, and likely other biogenic VOCs, shifts ozone production to the $NO_x$-sensitive regime for 8-9 hours of the day and surface ozone increases to 50-75 ppbv. A low pressure system moves in from the west at noon on 19 July, breaking up the stable high-pressure system (Kendon, 2022) and leading to cooler conditions on 20 July that decrease isoprene emissions, flatten diurnal variability in HCHO, and lessen the morning $NO_2$ peak. Diurnal variability in ozone, $NO_2$, HCHO and HCHO:$NO_2$ during the August heatwave (not shown)

mimics the July heatwave.

Single overpass instruments would diagnose ozone production during the heatwave in Central London as exhibiting weak sensitivity to $NO_x$ (HCHO:$NO_2$ ~ 2) for a morning overpass and very $NO_x$-sensitive (HCHO:$NO_2$ of 3-4) for an afternoon overpass. TROPOMI HCHO:$NO_2$ data are missing on two of the four heatwave days due to loss of HCHO data (Figure 5). On

days with TROPOMI data, HCHO:$NO_2$ replicates that from MAX-DOAS on 15, 16, and 20 August, but is slightly less than MAX-DOAS on 19 July (Figure 8(c)), despite consistent TROPOMI and MAX-DOAS HCHO and $NO_2$ on that day (Figure 5). The lower TROPOMI HCHO:$NO_2$ ratios in Figure 8 are due to differences in instrument sensitivity that cause greater decline in MAX-DOAS HCHO (48%) than in MAX-DOAS $NO_2$ (26%) in the comparison between TROPOMI and MAX-DOAS in Figure 5. Across the whole measurement period, mean TROPOMI HCHO:$NO_2$ is 45% more than MAX-DOAS,

suggestive that systematic differences in HCHO (TROPOMI > MAX-DOAS) and $NO_2$ (TROPOMI < MAX-DOAS) is greater than differences in vertical sensitivity of TROPOMI to HCHO and $NO_2$ (Section 3.2).

The UK air quality standard for surface ozone pollution is maximum daily 8-h average (MDA8) ozone of 100 μg m$^{-3}$ (equivalent to ~50 ppbv) not to be exceeded more than 10 times a year (DEFRA, 2022). We show in Figure 9 the time series

of MDA8 ozone averaged over the two urban background sites (Bloomsbury and North Kensington) from 1 January to 31 October 2022 (Section 2.4). For most of 2022, MDA8 ozone is below 80 μg m$^{-3}$. The standard is exceeded 18 times in 2022 and all exceedances are associated with increases in temperature. The three heatwaves account for most (67%) exceedances: four in June, three in July, and five in August. Site-to-site variability in the number of MDA8 ozone exceedances tracks



proximity to $NO_x$ sources from traffic. Marylebone Road, not included in the multisite mean MDA8 ozone due to very large

influence from local traffic, has one exceedance, whereas at each of the two sites included in the multisite mean MDA8 ozone

in Figure 9, exceedances total 15 at Bloomsbury and 24 at North Kensington. Fewer exceedances at the Bloomsbury site is

due to its proximity to a 2-lane congested road (Section 2.4).

In past heatwave years studied for ozone pollution episodes, MDA8 ozone exceedances averaged across all urban background

sites in Greater London in May-September totalled 17 in 1995, 19 in 1999, and 27 in 2003 (Doherty et al., 2009). This likely

includes a greater proportion of sites with less local traffic influence than the urban background sites in Figure 9. Isoprene

concentrations at Marylebone Road in the 2003 heatwave peaked at 1.6 ppbv (Lee et al., 2006), ~0.4 ppbv more than the peak

in Figure 8, though data are missing in summer 2022. Future increases in the number of ozone exceedances in Central London

is highly likely. The UK has committed to continued decline in national anthropogenic $NO_x$ emissions of 55% relative to 2005

values by any year in 2021-2029 and of 73% relative to 2005 by any year from 2030 (Office of the European Union, 2016)

and heatwaves are projected to increase in severity, frequency, and persistence due to climate change caused by anthropogenic

emissions of long-lived greenhouse gases (Christidis et al., 2020; Pörtner et al., 2022).

## 4 Conclusions

Here we report on ambient nitrogen dioxide ($NO_2$) and formaldehyde (HCHO) concentrations in Central London retrieved for

July-September 2022 from a recently installed MAX-DOAS instrument on the rooftop of an 11-story building at University

College London.

The high spatial resolution space-based TROPOMI instrument, capable of resolving sub-city atmospheric composition,

replicates day-to-day variability in MAX-DOAS $NO_2$ and HCHO (both with R = 0.71), but retrieves $NO_2$ columns that are 27-

31% less than MAX-DOAS and HCHO columns that are 20% more than MAX-DOAS.

Over Central London, clouds are detected in 60% of all observations using ratios of spectral intensity in the UV (330 nm) and

visible (404 nm). We find though that MAX-DOAS July-September mean retrievals obtained with and without clouds differ

by <3%, so MAX-DOAS offers complete day-time temporal coverage of UV/visible active components over frequently cloudy

Central London as long as most information in the retrieval is from the observations.

The $NO_2$ diurnal variability typically includes morning and evening peaks due to rush-hour traffic and a midday minimum

when photolytic loss of $NO_2$ dominates. HCHO diurnal variability is flat most days, except during heatwaves when warm

conditions increase temperature-dependent emissions of the biogenic VOC isoprene that oxidizes to form HCHO. On these

days, ozone production shifts from weakly $NO_x$-saturated in the early morning to strongly $NO_x$-limited at and around midday,



resulting in midday ozone concentrations of 50-75 ppbv. The regulatory standard of maximum daily 8-hour average (MDA8) ozone is exceeded 18 times in Central London in 2022; mostly during heatwaves.

Current single overpass space-based instruments observe Central London when $NO_2$ is at a midday minimum or undergoing steep decline following morning rush-hour, suffer data loss due to contamination by clouds and retrieval issues, and lead to different conclusions about ozone production regimes on heatwave days. The future Sentinel-4 geostationary instrument will address limited temporal sampling by observing Central London every daylight hour.

Continued emission controls targeting $NO_x$ sources and predicted climate change driven increases in heatwave occurrence, severity and longevity will inevitably increase ozone pollution episodes, necessitating continued reliance on forecasting and warning systems to mitigate harmful effects of heatwaves on public health.

**Code Availability.** The Python code of the RAPSODI retrieval algorithm can be requested from J-LT ([jan-lukas.tirpitz@airyx.de](mailto:jan-lukas.tirpitz@airyx.de)).

**Data Availability.** MAX-DOAS vertical profiles of $O_4$, $NO_2$ and HCHO during the measurement period are available for download in NetCDF format from the UCL Data Repository ([https://doi.org/10.5522/04/21610533](https://doi.org/10.5522/04/21610533)) (Marais et al., 2022).

**Author Contributions.** Study concept by RGR and EAM. Instrument installation by RGR and EAM with assistance from RR and JPM. J-LT and UF provided the RAPSODI code and guidance on its use. RGR and EG-S developed the cloud-flagging algorithm. EAM simulated GEOS-Chem. RGR led the data analysis. The manuscript was co-written by RGR and EAM. All authors reviewed and edited the manuscript.

**Competing Interests.** The authors declare they have no competing interests.

**Acknowledgements.** This research has been supported by the European Research Council under the European Union's Horizon 2020 research and innovation program (through the Starting Grant awarded to Eloise A. Marais, UpTrop (grant no. 851854)) and by the UCL Research Capital Infrastructure Fund 2021-22 awarded to Robert G. Ryan and Eloise A. Marais.

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

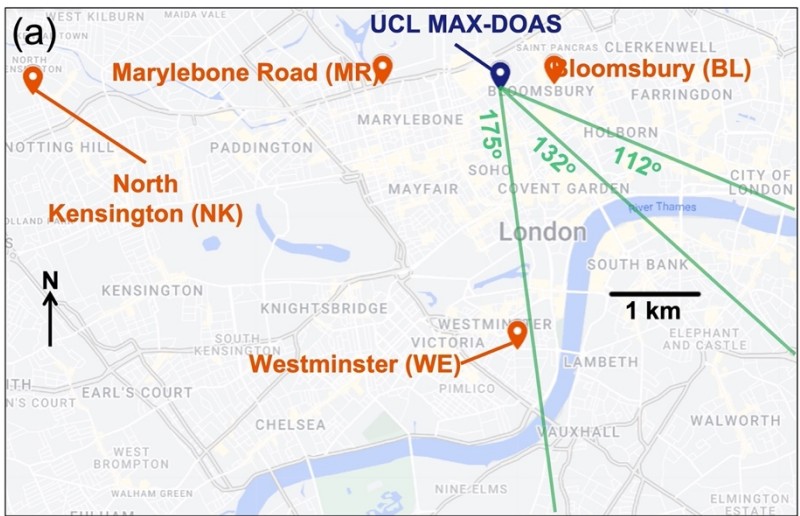
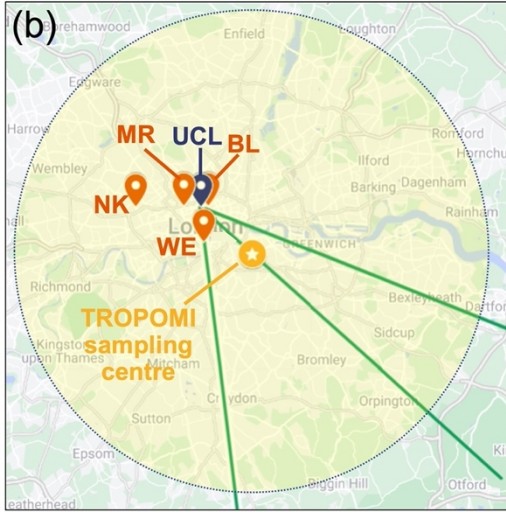

**Figure 1. Central London MAX-DOAS instrument location.** Maps show the MAX-DOAS site on the University College London (UCL) campus (a) and the centre and extent of coincident sampling of the TROPOMI instrument (b). Green lines in (a) and (b) are optimized viewing azimuth angles (see text for details). Red pins in (a) and (b) are nearby surface air quality monitoring sites. In (b), the orange filled circle is the TROPOMI sampling centre, the shaded yellow area the 20 km sampling radius of coincident TROPOMI pixels, and the purple line the 10 km visible horizon of the MAX-DOAS instrument (see text 720 for details). Maps from ©Google Maps 2022.

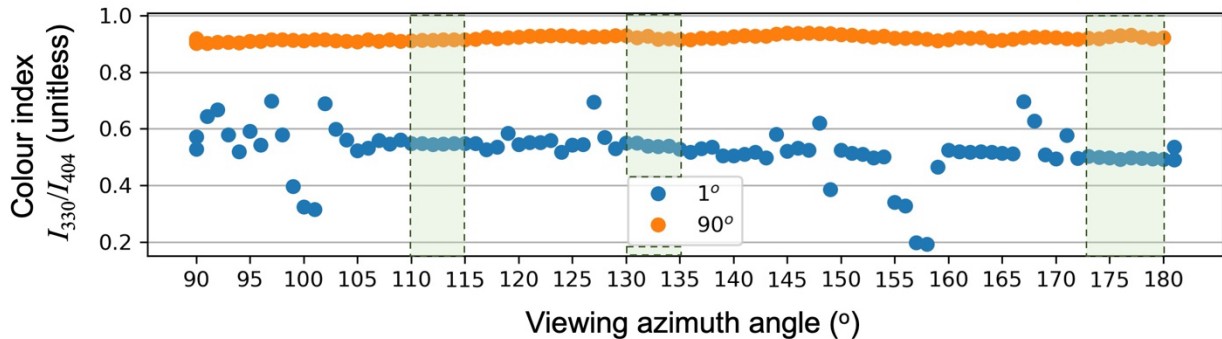




**Figure 2. MAX-DOAS colour index ($I_{330}/I_{404}$) measurements used for azimuth angle optimisation.** Intensity ratios at 1° (blue circles) and 90° (orange circles) elevation angles for a cloud-free horizon scan on 8 July 2022 at 14:15-15:00 UTC. Shaded green boxes indicate unobstructed regions.

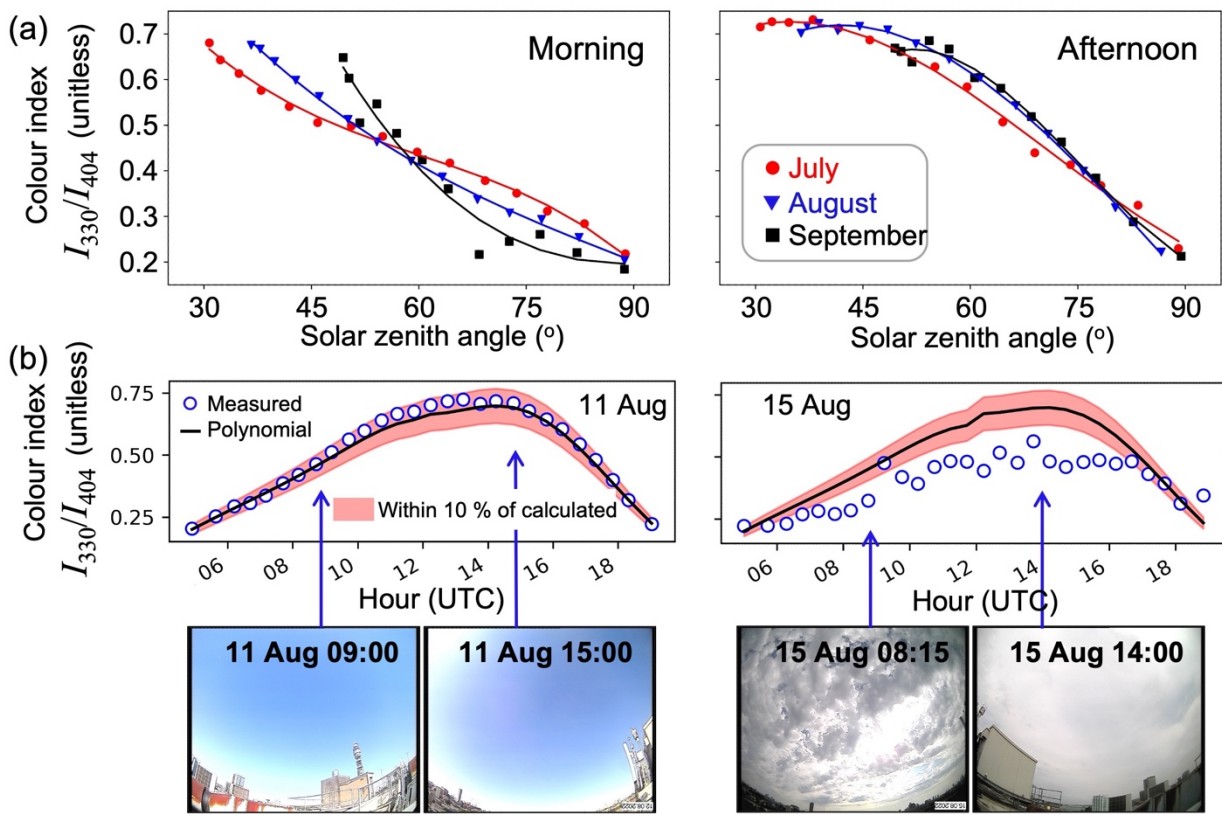


**Figure 3. Qualitative cloud detection.** Panels are morning (left) and afternoon (right) cloud-free colour indices ($I_{330}/I_{404}$) versus solar zenith angle (SZA) in July (red), August (blue) and September (black) (a), and diurnal variability of colour indices ($I_{330}/I_{404}$) on a cloud-free day and a cloudy day in August (b). Symbols in (a) are individual measurements and lines are 3rd order polynomial fits. Black lines in (b) are the cloud-free polynomials from (a) for August, peach shading indicates ±10% range from the fit, and blue open circles are individual measurements on 11th (left) and 15th (right) August. Inset camera images show representative sky conditions for each day.




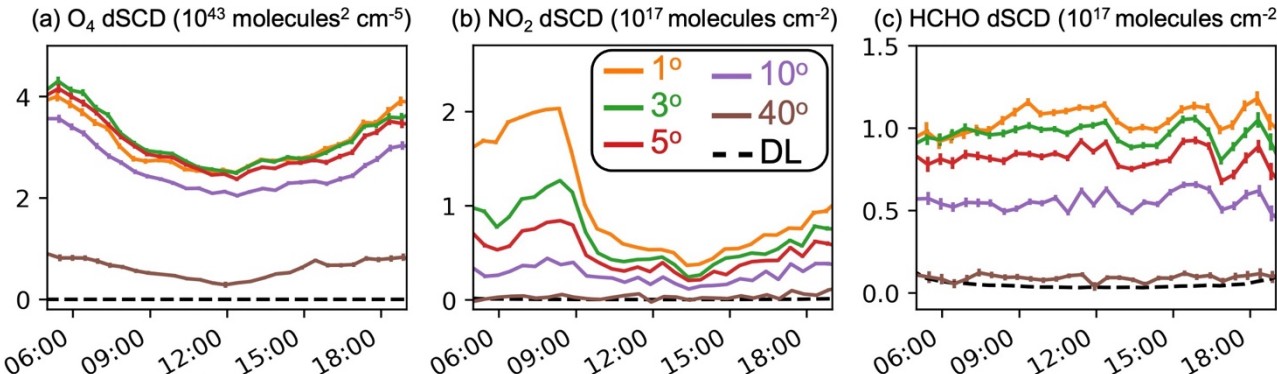


**Figure 4. Time series of O₄, NO₂ and HCHO differential slant column densities (dSCDs) on 18 July 2022.** DOASIS retrieved dSCDs of $O_4$ (a), $NO_2$ (b) and HCHO (c) at 1° (orange), 3° (green), 5° (red), 10° (mauve) and 40° (brown) elevation angles at the 132° azimuth angle (Figure 1). Error bars are dSCD uncertainties. Black dashed lines are detection limits (DL) at 1° elevation (see text for details).




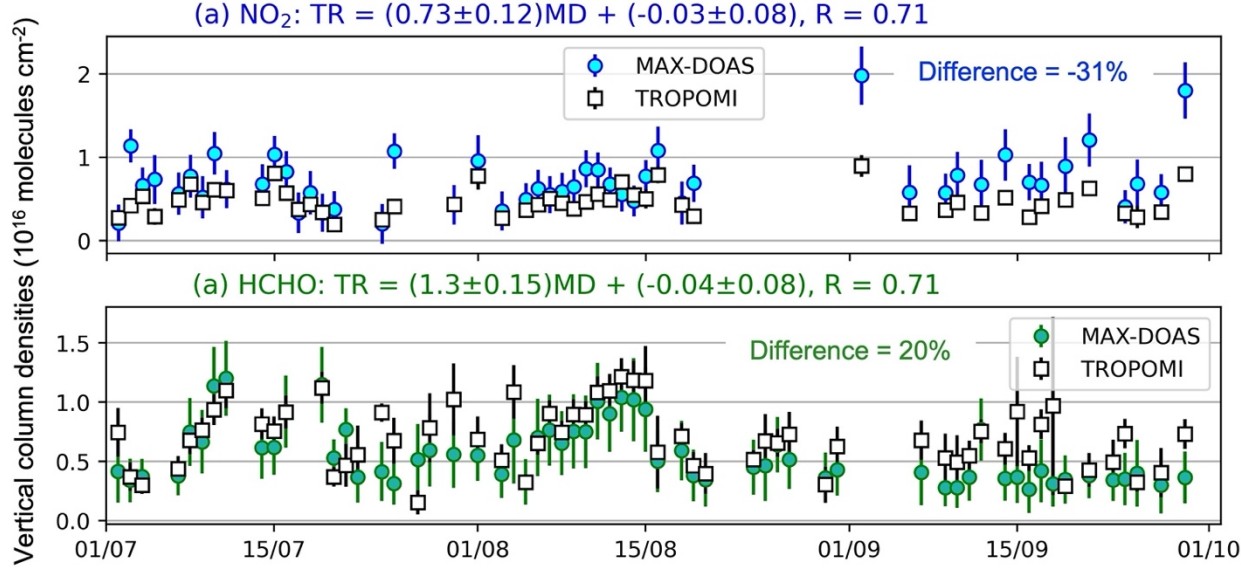

**Figure 5. Comparison of TROPOMI and MAX-DOAS over Central London in July-September 2022.** Panels compare coincident NO$_2$ (a) and HCHO (b) tropospheric vertical column densities from TROPOMI (open squares) and from MAX-DOAS smoothed with TROPOMI averaging kernels (filled circles). Error bars are retrieval uncertainties added in quadrature. Text above each panel gives the orthogonal distance regression of TROPOMI (TR) versus MAX-DOAS (MD), and the Pearson's correlation coefficient (R). Text in each panel is the median relative difference (TROPOMI minus MAX-DOAS). Heatwave periods are shaded orange.







**Figure 6. Comparison of daily daytime mean observations in July-September 2022.** Daytime data are those coincident with MAX-DOAS solar zenith angle < 90°. Panels are daily mean MAX-DOAS and surface site NO₂ (a), MAX-DOAS HCHO and surface site isoprene (b), MAX-DOAS HCHO:NO₂ (c), surface site ozone (d), and maximum air temperature at 60 m (e). MAX-DOAS values are the lowest retrieved layer in (a) and (b) and the vertical 0-8 km column in (c). MAX-DOAS results are the mean of all azimuth angles and surface sites are the mean of multiple sites for NO₂ and ozone, and of Marylebone Road only for isoprene (Section 2.4). Error bars are standard deviations of the multi-azimuth daytime means for MAX-DOAS, the site and daytime variability for in situ NO₂ and ozone, and daytime variability only for in situ isoprene. Inset values in (a) and (b) are Pearson's correlation coefficients (R). Shading shows heatwave periods (orange) in all panels and transition in ozone production regimes (grey) in (c). In situ daytime means are included if all sites have at least 4 hourly measurements that day.



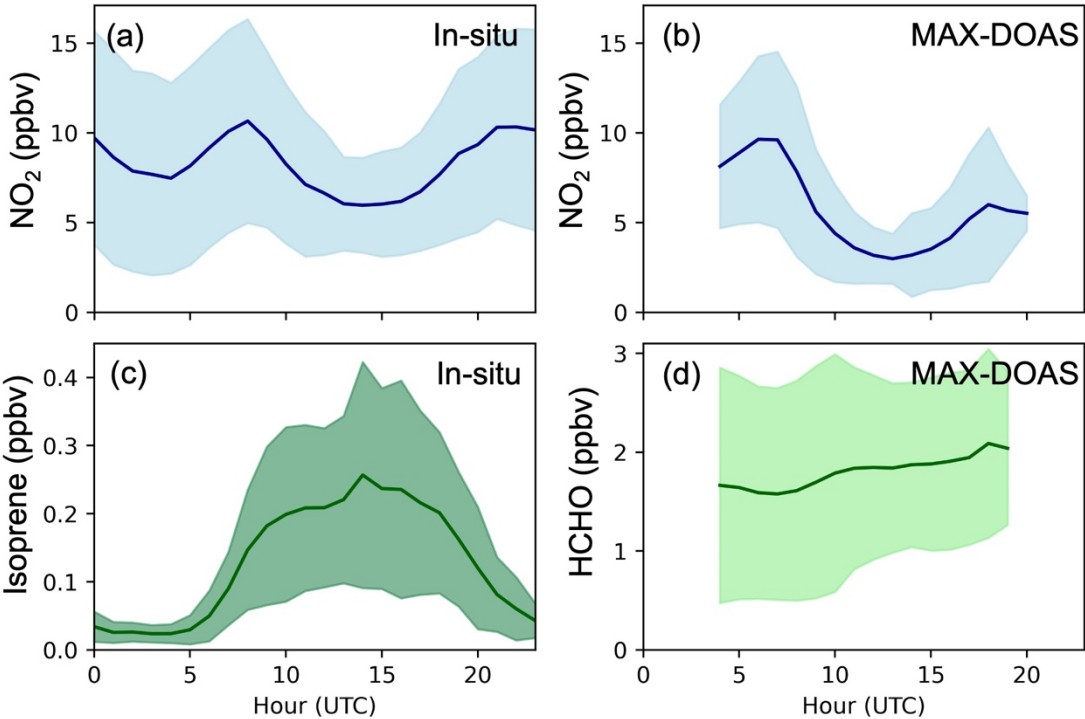


**Figure 7. Comparison of mean diurnal variation of NO₂, HCHO and isoprene averaged over July-September 2022.**
Columns are surface site NO$_2$ (a) and isoprene (c) and MAX-DOAS lowest retrieved layer NO$_2$ (b) and HCHO (d). Solid lines are means and shaded areas standard deviations of all July-September multi-site mean data in (a), Marylebone Road data in (c) and multi-azimuth angle mean data in (b) and (d).






**Figure 8. Comparison of hourly mean observations during the July heatwave.** Panels and features are the same as in Figure 6, except individual points are daytime (MAX-DOAS solar zenith angle < 90º) hourly means for 15-21 July 2022 and error bars are standard deviations of multi-azimuth hourly means for MAX-DOAS and site means for in situ $NO_2$ and ozone. Vertical lines show noon UTC as a guide. Additional data in (c) are collocated TROPOMI HCHO:$NO_2$ means (red crosses) and standard deviations (red error bars). MAX-DOAS and TROPOMI vertical sensitivities differ in (c).



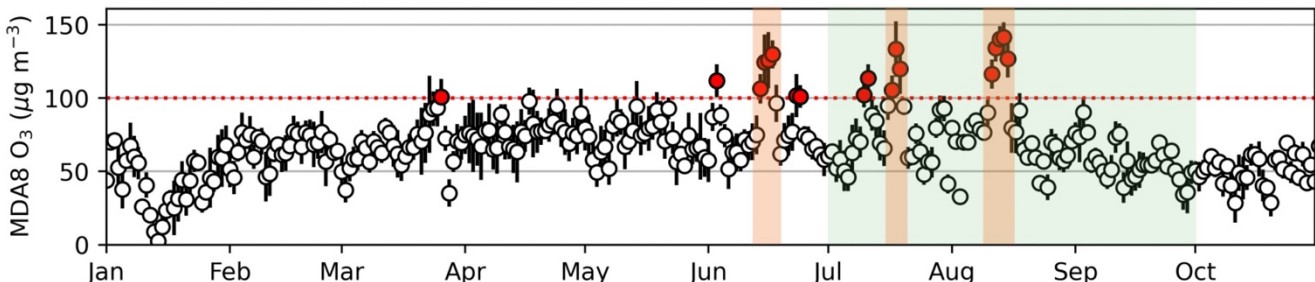

**Figure 9. Maximum daily 8-h average (MDA8) ozone in Central London from 1 January to 31 October 2022.** Points are MDA8 ozone values from multi-site mean hourly ozone (Section 2.4). Error bars are the multi-site standard deviations. The red horizontal dashed line is the UK standard of 100 µg m⁻³ (DEFRA, 2022). Points coloured red exceed the standard. Shading shows the 2022 heatwave periods (orange) and the MAX-DOAS measurement period (green).