# Peer review of "Measurement Report: MAX-DOAS measurements characterise Central London ozone pollution episodes during 2022 heatwaves"

_EGUsphere, 2023_

## Author Comment (AC1)

**POINT-BY-POINT RESPONSES**

Ms. Ref. No.: EGUSPHERE-2023-24, doi:10.5194/egusphere-2023-24. Title: Measurement Report: MAX-DOAS measurements characterise Central London ozone pollution episodes during 2022 heatwaves Journal: Atmos. Chem. Phys. Measurement Report

**To the Editor**

Point-by-point responses to reviewers are included below. Reviewer comments are in blue. Responses are in black and give line numbers consistent with the tracked changes manuscript. We have also applied very minor editorial changes to address typos identified on a careful reread of the manuscript and to edit 2 citations with updated page and issue numbers.

**Responses to RC#1:**

Ryan et al. provide an interesting and thorough report of the deployment of a MAX-DOAS instrument in central London and use its measurements to explore impact of the 2022 heatwaves on NO2 and HCHO concentrations, and subsequent effects on O3 production. The measurements are also compared with TROPOMI overpasses when available and highlight the complementary nature of the new measurements. The O3 production section is of wider interest in terms of urban air quality policy, as the authors note, with reference to the UK's goals of reducing NO2 emissions.

The paper is well written and formatted, and subject to some minor comments, should be accepted for publication in ACP.

**Minor comments:**

ca. figure 8 / lines 380 – the ratio between in-situ isoprene and HCHO appears to dramatically change during the 18th July. It would be interesting to have some discussion as the why this is, especially as the increase in HCHO happens before the increase in isoprene. I suspect it has something to do with the location of Marylebone Road relative to the MAX-DOAS view.

The reviewer comment prompted us to revisit the isoprene daytime mean and hourly data shown in Figures 6 and 8. Almost 24 hours of data are missing from 18 to 19 July that we now gap fill with values derived from the strong linear relationship between isoprene concentrations at Marylebone Road and London Eltham sites (R = 0.83). The orthogonal distance regression fit we use, for isoprene in ppbv, is: isopreneMarylebone = ( $0.23 \pm 0.004$ ) × isopreneEltham + ( $0.039 \pm 0.002$ ). This gap filling ensures that isoprene data shown in Figure 6 are more representative of daytime means and that we can assess temporal variability in isoprene/HCHO ratios in Figure 8. Updates to the manuscript include the methods detailing the approach we use to derive values when Marylebone Road observations are missing (Section 2.4, lines 270-274), values in Figures 6 and 8, analysis of the data in the manuscript (lines 412, 429-30, and 503-509), comparison to peak isoprene during the 2003 heatwave (line 582), and the Data Availability

statement that data on the UCL Data Repository also includes the gap filled isoprene data (https://doi.org/10.5522/04/21610533).

We now state that the large increase in HCHO on 18 July is likely due to accumulation of HCHO during very stagnant conditions and that daytime mean isoprene/HCHO values are the same during the period plotted in Figure 8 (updated figure pasted below). Ratio values are 89  $\pm$  43 pptv ppbv-1 on 15-17 July and 89  $\pm$  56 pptv ppbv-1 on 18-19 July. This suggests that the seemingly dramatic change in the ratio in the original manuscript is spurious. We also mention that HCHO yields from isoprene oxidation should be routinely high in London, due to sustained concentrations of NOx > 1 ppbv (lines 508-509).

---

## Author Comment (AC2)

**POINT-BY-POINT RESPONSES**

Ms. Ref. No.: EGUSPHERE-2023-24, doi:10.5194/egusphere-2023-24.
Title: Measurement Report: MAX-DOAS measurements characterise Central London ozone pollution episodes during 2022 heatwaves
Journal: Atmos. Chem. Phys. Measurement Report

**Responses to RC#1:**

*Ryan et al. provide an interesting and thorough report of the deployment of a MAX-DOAS instrument in central London and use its measurements to explore impact of the 2022 heatwaves on NO2 and HCHO concentrations, and subsequent effects on O3 production. The measurements are also compared with TROPOMI overpasses when available and highlight the complementary nature of the new measurements. The O3 production section is of wider interest in terms of urban air quality policy, as the authors note, with reference to the UK's goals of reducing NO2 emissions.*

*The paper is well written and formatted, and subject to some minor comments, should be accepted for publication in ACP.*

*Minor comments:*

*ca. figure 8 / lines 380 – the ratio between in-situ isoprene and HCHO appears to dramatically change during the 18th July. It would be interesting to have some discussion as the why this is, especially as the increase in HCHO happens before the increase in isoprene. I suspect it has something to do with the location of Marylebone Road relative to the MAX-DOAS view.*

The reviewer comment prompted us to revisit the isoprene daytime mean and hourly data shown in Figures 6 and 8. Almost 24 hours of data are missing from 18 to 19 July that we now gap fill with values derived from the strong linear relationship between isoprene concentrations at Marylebone Road and London Eltham sites ($R = 0.83$). The orthogonal distance regression fit we use, for isoprene in ppbv, is: $isoprene_{Marylebone} = (0.23 \pm 0.004) \times isoprene_{Eltham} + (0.039 \pm 0.002)$. This gap filling ensures that isoprene data shown in Figure 6 are more representative of daytime means and that we can assess temporal variability in isoprene/HCHO ratios in Figure 8. Updates to the manuscript include the methods detailing the approach we use to derive values when Marylebone Road observations are missing (Section 2.4, lines 270-274), values in Figures 6 and 8, analysis of the data in the manuscript (lines 412, 429-30, and 503-509), comparison to peak isoprene during the 2003 heatwave (line 582), and the Data Availability statement that data on the UCL Data Repository also includes the gap filled isoprene data (https://doi.org/10.5522/04/21610533).

We now state that the large increase in HCHO on 18 July is likely due to accumulation of HCHO during very stagnant conditions and that daytime mean isoprene/HCHO values are the same during the period plotted in Figure 8 (updated figure pasted below). Ratio values are 89 $\pm$ 43 pptv ppbv$^{-1}$ on 15-17 July and 89 $\pm$ 56 pptv ppbv$^{-1}$ on 18-19 July. This suggests that the

seemingly dramatic change in the ratio in the original manuscript is spurious. We also mention that HCHO yields from isoprene oxidation should be routinely high in London, due to sustained concentrations of $NO_x > 1$ ppbv (lines 508-509).

The diurnal profile of isoprene in Figure 7(c) is not affected by the additional gap-filled data.

[Figure]

**Figure 8. Comparison of hourly mean observations during the July heatwave.** Panels and features are the same as in Figure 6, except individual points are daytime (MAX-DOAS solar zenith angle < 90°) hourly means for 15-21 July 2022 and error bars are standard deviations of multi-azimuth hourly means for MAX-DOAS and site means for in situ $NO_2$ and ozone. Vertical lines show noon UTC as a guide. Additional data in (c) are collocated TROPOMI $HCHO:NO_2$ means (red crosses) and standard deviations (red error bars). MAX-DOAS and TROPOMI vertical sensitivities differ in (c).

*The lowest MAX-DOAS layer is described as ~55m, but the instrument is located at 60m above ground, and all elevation angles are listed as positive inclinations. Is this correct? Adding some information to section 2.1 for clarification would be useful.*

We have updated text in the manuscript to ensure it is clear that the vertical grid extends from 0 to 8 km above ground (line 147), that the instrument is located at 60 m and so is above the lowest retrieval layer (lines 149-150), and that negative viewing angles have been added to the

measurement sequence, but after the period of interest in this work (lines 148-149). We briefly discuss the influence of these on information content (DOFS) in the $NO_2$ vertical column (lines 317-319).

The reviewer comment also aided us in identifying that our initial interpretation of the RAPSODI grid as starting from the instrument altitude was not correct. Given that the lowest retrieval layer lies below the instrument altitude, we now estimate a near-surface concentrations of HCHO and $NO_2$ as the mean of the lowest 2 retrieval layers (lines 150-152). We use this to compare to the surface network in situ observations. In the comparison, we remove data with limited information from the observations, identified as DOFS summed over the lowest 2 retrieval layers less than 0.2 (lines 201-202). This has negligible effect on MAX-DOAS near-surface $NO_2$ and leads to very minor changes in comparison of MAX-DOAS near-surface HCHO and surface isoprene (Figures 6(b), 7(d) and 8(b)).

*Technical comments:*
*Figure 1 a – the labels collide with the location pins in some cases – adjust spacing to fix.*
Issue addressed. Updated figure pasted below. The caption is unchanged.

[Figure]

*The "DOFs/DOFS" initialism's "s" character is inconsistently capitalised throughout the manuscript.*
All are now consistently DOFS.

---

## Author Comment (AC3)

**POINT-BY-POINT RESPONSES**

Ms. Ref. No.: EGUSPHERE-2023-24, doi:10.5194/egusphere-2023-24.
Title: Measurement Report: MAX-DOAS measurements characterise Central London ozone pollution episodes during 2022 heatwaves
Journal: Atmos. Chem. Phys. Measurement Report

**Responses to RC#2:**

*Ryan et al. presented a valuable report about the ozone pollution episodes during 2022 heatwaves in Central London via the MAX-DOAS measurements. NO2 and HCHO VCDs of TROPOMI were firstly validated by ground-based MAX-DOAS. In addition, lowest layer retrieved NO2 and HCHO from MAX-DOAS were compared with in-situ NO2 and isoprene, respectively. Regarding the daytime ozone production, VOCs-limited regime is identified for non-heatwave days according to the MAX-DOAS HCHO-to-NO2 tropospheric vertical column ratios. Temperature favors the biogenic isoprene emissions and further the increase of ozone concentrations exceeding the regulatory standard. Influenced heavily by traffic emission, the compliance status may be changed under the conditions that stricter controls on NOx vehicle emissions and frequenter and severer heatwave. Overall, the paper is well organized and written, however, there still some comments need to be addressed before it can be considered to be accepted for ACP journal.*

*Main Concerns:*
*Since the vertical profiles of NO2 and HCHO can be obtained by the RAPSODI algorithm, why only the column density and lowest layer results were used to discuss in the paper? I would like to suggest the authors present the characteristics of the vertical pattern of NO2 and HCHO during heatwave days and non-heatwave days at least.*

Thank you for the suggestion. We chose to focus on the surface and the tropospheric column, as the goal of the study is to interpret surface ozone air pollution and assess tropospheric column density retrievals from the widely used TROPOMI instrument. Vertical profile information from MAX-DOAS is limited to ~3 pieces of information for $NO_2$ and ~2 for HCHO during the TROPOMI overpass, as stated in the manuscript when reporting on typical DOFS (line 317).

*Moreover, considering the air mass transport described in Line 48-49, the HCHO-to-NO2 ratio may also be analyzed in different heights.*

This is surface air advected from continental Europe. We now state "surface" to ensure this is clear (line 56).

*Minors:*

*1. Line 45-47, It's better to also provide background information about the VOCs emission in Central London, when discuss the ozone production there.*

We now include a brief discussion of the mix of summertime VOCs sources in London measured during a field campaign in London and postulate that volatile chemical products (VCPs) likely also contribute to enhancements in emissions in the morning (lines 51-54).

*2. Line 115-120, in addition to the DLs of individual DSCDs, the authors should provide a more detailed table for the spectral analysis configurations. Besides, the performance of the spectral analysis should be evaluated, such as the range of RMS? DSCDs errors? An example plot of spectral fitting? And any filtering of the DSCDs before be introduced into the profile retrieval?*

We now include Table 1 (screenshot pasted below) to summarize additional fit parameters not given in the text (Section 2.2, lines 130-134). The lineshapes for HCHO, $NO_2$ and $O_4$ are well documented in the MAX-DOAS literature in the spectral fitting example plots, so instead of reproducing these, we evaluate the spectral analysis by presenting the mean dSCD errors (<5 % for all trace gases, line 292) and the mean residual RMS of $4 \times 10^{-4}$ for all fitting windows (line 299); corrected from $4 \times 10^{-5}$ in the original manuscript. No other filtering is applied to the dSCDs.

**Table 1.** Cross section fittings for retrieving $NO_2$, HCHO, and $O_4$ dSCDs.

| Absorber | Temperature [K] [a] | $I_o$ correction [molecules cm$^{-2}$] [b] | Reference |
|---|---|---|---|
| $NO_2$ | 220, 294 | $1 \times 10^{15}$ | Vandaele et al. (1998) |
| HCHO | 298 | $5 \times 10^{15}$ | Chance and Orphal (2011) |
| $O_4$ | 293 | $3 \times 10^{43}$ | Finkenzeller and Volkamer (2022) |
| $O_3$ | 223, 246, 293 | $1 \times 10^{18}$ | Serdyuchenko et al. (2014) |
| BrO [c] | 223 | $1 \times 10^{13}$ | Fleischmann et al. (2004) |

[a] Temperatures used for cross section fit. More than one temperature given for trace gases with substantial contribution from the warm troposphere, cold upper troposphere and stratosphere, [b] Solar reference intensity ($I_o$) correction, [c] Bromine monoxide.

*3. Fig. 4, why datasets of elevation 20° not be presented?*

Data for the 2° elevation angle were also not shown. This was merely to avoid showing a cluttered figure. We now show in Figure 4 all elevation angles. On updating the plot, we also identified a plotting issue with the $NO_2$ dSCDs that has been addressed. Updated plot and caption pasted below.

[Figure]

**Figure 4. Time series of O₄, NO₂ and HCHO differential slant column densities (dSCDs) on 18 July 2022.** DOASIS retrieved dSCDs of $O_4$ (a), $NO_2$ (b) and HCHO (c) at 1° (blue), 2° (orange), 3° (green), 5° (red), 10° (mauve), 20° (brown) and 40° (pink) elevation angles at the 132° azimuth angle (Figure 1). Error bars are dSCD uncertainties. Black dashed lines are detection limits (*DL*) at 1° elevation (see text for details).

*4. Fig. 6, MAX-DOAS HCHO:NO2 is VCD to VCD or lowest layer to lower layer? Same comments on Fig. 8.*

It already states in the legend for panel (c) that this is the ratio of the columns. For clarity, we now add "vertical column density ratios" in "MAX-DOAS HCHO:NO₂ vertical column density ratios (c)" to the Figure 6 caption. Figure 8 also already states that the panels in Figure 8 are the same as those in Figure 6.

*5. Fig. 8 and related discussion, the dependency of isoprene-to-HCHO ratio to NO2 need to be investigated, also isoprene-to-NO2 ratio.*

We assume the reviewer is referring to the dependence of HCHO yields from isoprene oxidation on $NO_x$ concentrations. In London, the $NO_x$ concentrations far exceed the threshold between low- and high-$NO_x$ oxidation conditions (~1 ppbv; Marais et al., ACP, doi:10.5194/acp-12-6219-2012, 2012), due to sustained large emissions of $NO_x$ from vehicles. As such, we do not expect dependence of HCHO yields from isoprene oxidation on $NO_x$ in London in 2022. We now clarify this is the case in the text (lines 508-509).